# Development of a Prototype for a Bilingual Patient-Reported Outcome Measure of the Important Health Aspects of Quality of Life in People Living with HIV: The Preference Based HIV Index (PB-HIV)

**DOI:** 10.3390/jpm12122080

**Published:** 2022-12-16

**Authors:** Kedar K. V. Mate, Bertrand Lebouché, Marie-Josée Brouillette, Lesley K. Fellows, Nancy E. Mayo

**Affiliations:** 1Department of Family Medicine, Faculty of Medicine and Health Sciences, McGill University, Montreal, QC H3S 1Z1, Canada; 2Centre for Health Outcomes Research and Evaluation, Research Institute of the McGill University Health Centre, Montreal, QC H4A 3S5, Canada; 3Department of Psychiatry, McGill University Health Centre, Montreal, QC H4A 3J1, Canada; 4Department of Neurology and Neurosurgery, Montreal Neurological Institute, McGill University, Montreal, QC H3A 2B4, Canada; 5School of Physical and Occupational Therapy, McGill University, Montreal, QC H3G 1Y5, Canada; 6Divisions of Clinical Epidemiology, Geriatrics, Experimental Medicine, Department of Medicine and Health Sciences, McGill University, Montreal, QC H4A 3J1, Canada

**Keywords:** preference-based measure, health-related quality of life, HIV, patient-reported outcome measure, patient-generated index

## Abstract

(1) Background: The aim of this project was to develop a short, HIV-specific, health-related quality of life measure with a scoring system based on patient preferences for the different dimensions of the Preference-Based HIV Index (PB-HIV). (2) Methods: This study is a cross-sectional analysis of data from the Canadian Positive Brain Health Now cohort (n = 854; mean age 53 years). Items from the standardized measures were mapped to the areas from the Patient-Generated Index and formed the domains. A Rasch analysis was used to identify the best performing item to represent each dimension. Each item was then regressed on self-rated health (scored 0 to 100) and the regression parameters were used as scaling weights to form an index score for the prototype measure. (3) Results: Seven independent dimensions with three declarative statements ordered as response options formed the PB-HIV Index (pain, fatigue, memory/concentration, sleep, physical appearance/body image, depression, motivation). Regression parameters from a multivariable model yielded a measure with a scoring range from 0 (worst health) to 100 (perfect health). (4) Conclusions: Preference-based measures are optimal, as the total score reflects gains in some dimensions balanced against losses in others. The PB-HIV Index is the first HIV-specific preference-based measure.

## 1. Introduction

Quality of Life (QOL) [1] is defined by the World Health Organization as “individuals’ perception of their position in life in the context of the culture in which they live and in relation to their goals, expectations, standards and concerns”. In the context of living with a health condition such as HIV, QOL goes beyond a description of health status, also reflecting the way that people perceive and react to their health status and to other, nonmedical aspects of their lives [2]. Health-related quality of life (HRQL) measures covering the most common health aspects of quality of life have become an integral part of clinical research. The first reference to the term HRQL or “Well-Years” appears in a 1982 is in a publication by Kaplan and Bush in reference to the output of health program evaluation [3]. Since then, HRQL has been studied extensively in almost all health conditions and used for evaluations of all types of health interventions, including those preventive, curative, rehabilitative, and palliative. Generic HRQL measures are designed to be used in the general population and across health conditions. Disease-specific HRQL measures have been developed for almost all health conditions. A challenge with most measures of HRQL is that they are of the profile type, in that there are multiple domains for each with multiple items, leading to multiple scores [4]. The items themselves do not function well on their own; instead they are summed to yield multiple domain-specific scores. There are disadvantages to having a HRQL outcomes represented by multiple domains, as associations with the exposures or treatments under consideration may be differently affected across domains, making interpretations difficult [4]. However, in order for a single score to yield meaningful information, a weighting system for the domains needs to be used. The methods to create a mathematically sound score from multiple items/domains are complex, as each domain may not contribute equally to the construct value [4]. Weights can be derived mathematically, using a method such as principal component analysis, statistically by using impact weights that are usually based on mortality or incidence data [5,6,7,8], or by incorporating patient preferences [9,10,11,12]. Several generic preference-based measures are reported in the literature, the most widely used being the EuroQol-5 dimensions (EQ-5D), the Health Utilities Index Mark 2 and 3 (HUI 2 and 3), and the Short Form-6 dimension (SF-6D). These use preferences derived from the general population as weights because the aim is to compare across conditions and the weights consider that society ultimately pays for interventions. Preference-based measures are unique in that they have one item per domain; this “best” item represents each dimension, and the dimensions are independent of each other. A patient-centered approach to measurement would use patient preferences as the weighting system. This approach has been used to create HRQL measures for a number of conditions [13,14,15,16,17].

Improvement in combined antiretroviral therapy (cART) has led to increased life expectancies of people living with HIV (PLWH) [18]. However, HIV requires lifelong follow-up, self-management, and antiretroviral medication adherence [19,20]. It is a complex chronic condition affecting a population that often faces multiple psychosocial disadvantages, comorbidities, and psycho-behavioral problems, including those associated with aging [21,22]. Despite the improvements in controlling the infection, the condition still has a notable negative impact on health-related quality of life (HRQL), even in people who are virally suppressed on combination ART (cART) [23].

In the context of HIV, only profile HRQL measures are available. These are long and are rarely used clinically. As people living with HIV need to make treatment decisions that can affect various aspects of health-related quality of life differently, a preference-based measure would fill a gap for an outcome measure for both clinical care and research. It is also possible that different health aspects of quality of life also affect the decision to seek different modes of health care delivery and thus the dimensions of an HIV-specific preference-based measure could also serve to quantify a person’s propensity to accept therapeutic options or choose interventions to improve health-related quality of life such as rehabilitation or self-management. As accepting therapy options or recommendations is a behavior, a behavior change model would seem ideal for structuring the content of this new measure.

The global aim of this study is to estimate the extent to which developing a short, HIV-specific, theory-informed HRQL measure, eventually with the different dimensions to be weighted based on patient preferences, is feasible and yields values that correspond to values from generic and HIV-specific HRQL measures. The hypotheses were: (i) the prototype measure would relate moderately to converging constructs; and (ii) the measure would behave as expected across groups known to differ on the constructs. Estimates of feasibility and relationships with other measures will be derived from a prototype measure. This experience is needed before proceeding with the preference weighting of the dimensions to form a final version.

## 2. Materials and Methods

The methods to develop the Preference-Based HIV HRQL measure followed the guidelines recommended by the Food and Drug Administration [24,25] for developing a patient-reported outcome measure (PROM) and methods used to develop other such preference-based measures [13,26]. Briefly, these methods recommend a strong conceptual framework, gathering input directly from patients on content, items, time frames and response options, conducting studies which test the behavior of the new items, and the development of a strong scoring system for a total score.

This paper reports on the prototype phase to provide evidence, using existing data, that a short multi-dimensional index will behave comparably to existing generic measures or longer HIV-specific measures and thus support moving forward through the other steps of development. Figure 1 outlines the steps used to develop the prototype measure.

This is the correct figure (delete this text).

Data source: The data for this analysis was obtained from entry evaluation of participants in the Positive Brain Health Now (+BHN) cohort (https://brainhealthnow.org; accessed on 27 January 2021), which has been described in numerous publications [27,28,29,30,31,32,33], all of which were used to develop the conceptual framework and content for the new measure. Briefly, the +BHN cohort comprises an initial sample of 856 HIV-positive men and women over the age of 35 years who were recruited through consecutive sampling at HIV clinics in four Canadian cities: Montreal, Toronto, Hamilton, and Vancouver. The vast majority of participants were taking antiretroviral therapy; the majority had achieved viral suppression. The project was approved by institutional review boards at all sites and performed in accordance with the Declaration of Helsinki. All participants provided written informed consent.

The strength of this cohort is that members are fully characterized on patient-centered outcomes based on a strong theoretical framework, the Wilson-Cleary model, and widely tested and used in the health outcomes research [27]. While all consent-based observational studies have a potential for selection bias, the +BHN cohort was able to estimate the direction and impact of such bias [32], which resulted in the recruitment of a more vulnerable sample of people.

+BHN Platform Measures: Participants filled out a total of 17 questionnaires in addition to providing information on sociodemographic variables and completing a battery of neurocognitive tests [27,32,34,35].

A unique feature of the measurement plan was the inclusion of an individualized measure, the Patient-Generated Index (PGI) [36]. The PGI is a widely used individualized QOL measure that queries people to specify up to five areas of their lives which are affected by their health condition. The nominated areas have been previously reported and include: cognition, fatigue, emotional function, stigma, perception of self and body, exercise tolerance, work, recreation, relationships, intimacy, and health [33].

Step 1, 2, and 3 QOL domains and item mapping: The areas indicated by the people living with HIV were obtained from the PGI that completed the first two steps of the methods shown in Figure 1. The text threads from the PGI were mapped to the World Health Organization’s International Classification of Functioning [37].

In this context, domains represent broad latent constructs, and the items are single questions. Items from the platform measures were mapped to the functional domains. The available measures, fully described in previous papers [27,30], were RAND-36 [38], three items from Starkstein’s Apathy Scale [39], the Older Americans Resources and Services Social Support (OARS) [40], the Life Engagement, Hospital Anxiety and Depression Scale [41], the WHOQOL-BREF [1], the WHO-5 Well-Being Index [42], the Perceived Deficit Questionnaire (PDQ) [43], the Pittsburgh Sleep Quality Index [44], and the Trier Inventory for Chronic Stress [45]. Other variables of interest for interpretation of the new measure were age, sex, education, time since HIV diagnosis, nadir CD4, and self-reported medication adherence [46,47].

Step 4. Rasch analysis: A Rasch analysis was conducted on all reflective domains and one ‘best’ performing item that covered the latent trait was selected to represent the dimension. The Rasch analysis was performed using the Rasch Unidimensional Measurement Model (RUMM2030).

Step 5 and 6. Item selection: Dimension independence is a requirement of a multidimensional health state measure. Only one item per domain that covered the underlying latent trait was selected to represent a dimension. Item-to-item correlations were performed to eliminate correlated dimensions (r ≥ 0.6).

Step 7. Establish scaling values: EQ-VAS is a vertical visual analog scale of self-rated perception of health with endpoints labelled as ‘the best health you can imagine’ and ‘the worst health you can imagine’. EQ-VAS was regressed separately on each dimension with the levels represented as separate categories. The scaling process mimics weights derived using rating scale methods [48,49]. Level 1 (no problem) was selected as the reference to estimate the health impact of greater problems. The unstandardized regression coefficients were used as scaling values, rescaled to add to 100.

Step 8. Interpretability: This was assessed using the magnitude of the correlation with convergent constructs (see measurement section) and of the differences across known-groups [50] defined by categories of age, year of diagnosis (pre- or post- combined ART era–1996), nadir CD4, comorbidity, and medication adherence. As correlations were among the measures representing the same latent construct, correlations of ≥0.8 are considered strong and 0.5 to 0.8 are considered moderate [50]. EQ utility score, EQ-VAS and PB-HIV Index were regressed on self-reported adherence to anti-retroviral medications. Pearson’s correlations with 95% confidence interval (95% CI) were computed between two variables with continuous measurement scale. All statistical analysis were performed using the Statistical Analysis Systems (SAS, version 9.4).

Step 9. Focus groups to formulate items in two languages: The process for formulating the items during the focus groups followed that described of developing a measure for people with Multiple Sclerosis [51]. The author BL has an active collaboration with the Community Advisory Committee, which is made up of volunteers people living with HIV. The committee was invited to participate in the focus group. To participate in the focus group the patients had to be over 18 years, diagnosed with HIV for at least a year, taking cART, and have no self-reported cognitive impairment. At least one participant had to be female, and at least two had to be fluently bilingual in English and French. Healthcare professionals were identified and invited through the network of co-authors. To participate, the healthcare professional had to have some experience with patient interaction and either be a licensed professional or in be training.

Two separate focus groups were conducted, the first with the healthcare professionals and the second the patient experts. As there were only seven items, only one focus group was conducted with each group. The participants were provided with the item list a week in advance, in order to think and reflect. The focus groups were conducted online over Zoom by the first author, KM, and were voice recorded. At the outset, the interviewer gave a short presentation (10 min) which outlined the motivation for creating the measure and the steps used to measure development. Following the presentation, each item was presented in English and French, and the participants were invited to discuss them. The interviewer did not provide leading questions but provided clarifications as needed. The focus group took approximately an hour and based on the feedback from the interaction the revised items were shared with the whole focus group. The participants were asked the following questions while reading through the items: (1) Recall period: What do you think should be the recall period: today, over the past week, and over the past four weeks? Why did you choose this recall period?; (2) In your own words, what are the statements referring to?; (3) How are the three declarative response statements different?; (4) Are these statements easy to understand?; (5) Are any words in these statements unclear?; and (6) How could we improve the wording?

The original items were written in English and parallel or simultaneous translation was conducted in French so that the interpretation of the item between the two languages was concordant. If there was any word or phrase that could not be translated to French, it was abandoned in the English version of the measure and an alternative was suggested by the focus group.

## 3. Results

Table 1 presents the sociodemographic characteristics of the people who were included in the +BHN Cohort.

At baseline, data were available on 728 men and 136 women with a mean (standard deviation—SD) age of 53.4 (8.3) years and 50.7 (7.4) years, respectively. Over 90% of the participants were above the level of high school graduate. All participants in the cohort were on cART. A total of 810 people completed the PGI, and 3044 text threads were mapped to the ICF, yielded 34 domains. The most prevalent nominated areas were health (97%), emotional function, intimacy, work/school, relationships, recreation/leisure, stigma, perception of self/body image, cognition, exercise tolerance, and fatigue. The domains work/school, relationship, and recreation/leisure were excluded from the PB-HIV index as these domains are not independent; in other words, these domains depend on other aspects of life such as physical function or cognition. Pain was nominated by people with HIV, but not often; however, it was included for content coverage. There was only one item on intimacy, asking participants “how satisfied they were with their sex life”, which was excluded from the analysis as being outside of the health domain. Exercise tolerance was correlated with fatigue and the usual activities; of these items, fatigue was retained. After eliminating the dependent domains of participation and health, correlated items within the domains of emotional function and cognition, one non-health domain (intimacy), and seven independent dimensions remained.

Table 2 shows the original items and their corresponding scales for seven dimensions with unstandardized regression coefficients and scaling values.

The seven dimensions are pain/discomfort, fatigue, memory/concentration, physical appearance/body image, sleep, depression, and motivation. To illustrate, the item on pain/discomfort originated from the EQ-5D-3L, and had three levels with the referent ‘I rarely have pain or discomfort’ that is weighted at zero. For the next two levels, ‘I have pain or discomfort some days’, and ‘I have pain or discomfort most days’, the regression parameters (SE) are 9.4 (2.1) and 13.9 (2.2) and the corresponding weights are 9 and 14. The fatigue item from RAND-36 has a total of six response options. These six responses were grouped together to create a three-level fatigue item. The responses ‘a little of the time’ and ‘none of the time’ were combined and used as a referent (Level 0). Similarly, ‘some of the time’ and ‘a good bit of the time’ were combined for Level 1 and ‘most of the time’ and ‘all of the time’ were combined for Level 2. All regression parameter estimates showed monotonicity, in other words, the beta estimates increased from level 1 to level 2. Three items: exercise tolerance (RAND-36), enjoy life (WHOQOL-BREF), and usual activities (EQ-5D-3L) were highly correlated and therefor deleted (r ≥ 0.5). One item on social support (OARS) was eliminated, as the item fell outside the purview of the health care system.

Table 3 shows the mean and correlations with 95% CI between the PB-HIV Index and other converging constructs.

Column 3 shows that the highest correlation (r = 0.78, 95%CI = 0.75 to 0.81) was between PH-HIV Index total score and psychological domain from WHOQOL-BREF, whereas the smallest correlation (r = 0.14, 95% CI= 0.07 to 0.21) was between Index total score and the Brief-Cognitive Ability Measure, a composite of several computerized cognitive tests (B-CAM).

Table 4 presents item-to-item polychoric correlations of the PB-HIV Index.

The correlation coefficients ranged from 0.16 between the motivation and pain dimensions to 0.55 between the sleep and fatigue dimensions. The highest correlations were seen between fatigue and sleep dimensions and the lowest correlations were present memory and motivation dimensions.

Table 5 shows the results of logistic regression on self-report adherence to medications regressed on the EQ-VAS and PB-HIV Index.

For every 10 units difference on PB-HIV, the odds of forgetting to take medications is higher by a factor of 1.13.

Table 6 presents evidence that the PB-HIV Index and other generic measures of disease severity behaved as expected across sex, age, before or after cART (1996), nadir CD4 cell count, and comorbidities.

The mean [52] PB-HIV Index score for men was 65.2 (20.7) and for women it was 60.5 (24.5). These scores for the SF Index were 69.4 (12.4) and 69.2 (12.5) and for EQ Utility they were 81.5 (16.4) and 80.8 (17.4). The PB-HIV Index had a wider range of values (5 versus 1) on SF Index and (5 versus 1) on EQ Utility. This pattern was more or less consistently observed between the PB-HIV Index and other measures.

Table 7 shows the characteristics of patient experts and healthcare professionals who participated in focus groups and simultaneous translation.

A total of 13 participants, five patient experts and eight healthcare professionals, participated in the translation exercise. There was a mix of ethnicity, a range of educational profiles, and bilingualism among the participants.

The original items served to construct a prototype, but these items needed to be rewritten to meet the needs of a preference-based measure and a bilingual population. Item writing was done simultaneously in English and French and cognitive reflection on the wording and response options was done at the same time. Table 8 shows the PB-HIV Index with seven dimensions with three declarative statements in both English and French.

The prototype Index consists of seven dimensions and three levels with a total of 2187 possible health states. A person living with HIV who has no problem on any of the seven dimensions will have a health state of 1111111 and a score of 100. Figure 2 shows the distribution of PH-HIV Index total scores with a mean of 64.4 and SD of 21.49.

## 4. Discussion

The present study reported on the development of a short prototype preference-based HRQL Index (PB-HIV Index) for people living with HIV. The results showed that this approach yielded a measure that performed as well as generic measures (RAND-36, and EQ-5D-3L) and domains from an HIV-specific measure (WHOQOL-HIV BREF).

The resultant prototype described here is now ready for the development of a scoring algorithm. Our approach will be to weight the dimensions according to the preferences of people living with HIV. This approach is different from how the EQ-5D-3L is weighted where preferences for the different health states are obtained from the general population and not from people living with the health condition being evaluated [53,54,55,56].

This patient-preference approach also differs from how condition-specific HRQL measures are scored. HIV-specific measures such as the WHOQOL-HIV-BREF (31 items) are of the profile type with one score per domain (physical health; psychological health; level of independence; social relationship; environmental health; and spirituality, religion, and personal beliefs). Many of these domains are outside of the influence of the health care system. A newer measure, PROQOL, covers 11 domains: general health perception, social relationships, emotions, energy/fatigue, sleep, cognitive functioning, physical and daily activity, coping, future, symptoms, and treatment across nine countries [57]. More of these domains are actionable in the context of healthcare, but there is no total score and, again, a total of 30 items need to be administered and scored. In general, the use of profile measures makes interpretations across domains difficult when some change and others do not.

Our protype PB-HIV Index, with only seven dimensions covering the important domains included in other longer measures, shows strong potential for being feasible to include in both clinical settings and research.

Furthermore, the protype PB-HIV Index predicted adherence to cART in people living with HIV (see Table 5). This is not surprising given that the dimensions could themselves serve as a barrier to adherence. Thus, the PB-HIV Index could not only be an outcome for health interventions but also a propensity indicator for adherence to cART.

The PB-HIV Index had two dimensions, pain/discomfort and depression, that are also captured by generic EQ-5D, and one dimension, pain, in common with the SF-6D measure. The dimension of physical appearance/body image is unique to this population and is not captured by any generic quality of life measure. HIV-associated lipodystrophy syndrome and fat redistribution is induced by certain antiretroviral medications. The resulting disfigurement is a barrier to long-term adherence to ART, and leads people living with HIV to switch to other medications [58].

Disease-specific preference-based measures are becoming more popular because they not only include areas of health that are important to the population, but they also are scored based on how important each dimension is to the population. As such, they are more likely to detect change, for example following changes in medication or lifestyle.

Generic preference-based measures provide health states and index scores that are linked to quality-adjusted life years (QALYs), a measure that values the years of life remaining following treatment or intervention which is scored on a scale of 0 to 1 where 0 is death, and 1 is perfect health [59,60,61,62]. The weight for each dimension comes from members of the general public, as they are considered neutral with respect to any one health condition. This allows for treatment options to be compared across conditions, and this is often used to allocate scarce resources.

Condition-specific preference-based measures, on the other hand, are used primarily for the comparative effectiveness of therapies applied to people with the condition. The perspective is that the people living with the health condition are the best judges of how significant the health dimensions are to their quality of life [63]. HIV is now a covert health condition, and people with HIV are living fulfilling lives without manifest physical disabilities, as was the case in the past. However, the data show that the population experiences fatigue, poor sleep, depression, lack of motivation, and cognitive deficits, which are considered to be “hidden disabilities”. The impact of these hidden disabilities may not be fully appreciated by members of the general population [63], and hence including the patient’s voice should be considered important.

Strengths and limitations

The development of preference-based measures in different health conditions is a growing area of research [9,10,13,51]. The availability of data from PGI, which is a semiqualitative individualized measure, was an advantage for the development of the PB-HIV Index. The dataset from the +BHN cohort was well-suited for the present purposes because the measurement framework was comprehensive and was based on a strong theoretical model [64].

One limitation to the current scoring algorithm was that the reference for assigning weights for each dimension was based on regressing self-rated health measured by the EQ-VAS on each dimension. Although the EQ-VAS relates to current health, it may be that people find it difficult to separate their ‘health’ from other aspects of life [65].

Preference-based measures typically only have a few dimensions, as the number of unique health states is a function of the number of response options raised to the power of dimensions. As such, the PB-HIV Index would have 3^7^ or 2,147 unique health states that would need to be valued. Knowing this at onset, decisions had to be made about which of the original 13 dimensions should be prioritized for inclusion. Thus, we focused on those dimensions most directly under the influence of the health care system. The six domains (walking, apathy, stigma, intimacy, relationship, recreation/leisure) not included had less impact on the health rating used to derive weights (Appendix A).

## 5. Conclusions

The final PB-HIV Index consists of seven dimensions (pain/discomfort, fatigue, memory/concentration, physical appearance/body image, sleep, and anxiety/depression, motivation) evaluated on a three-point scale, making this approach to the evaluation of HRQL feasible for research and clinical practice owing to the low response burden. The dimensions were appraised by stakeholders from the HIV community who guided the wording of the items, in both English and French, to create a set of items that captured the desired meaning. Additional cognitive debriefing on a more diverse sample is warranted before preference weights are derived. The final version will need to be tested in different sub-groups of people living with HIV to provide evidence that this new measure produces meaningful and interpretable data supporting its use.

## Figures and Tables

**Figure 1 jpm-12-02080-f001:**
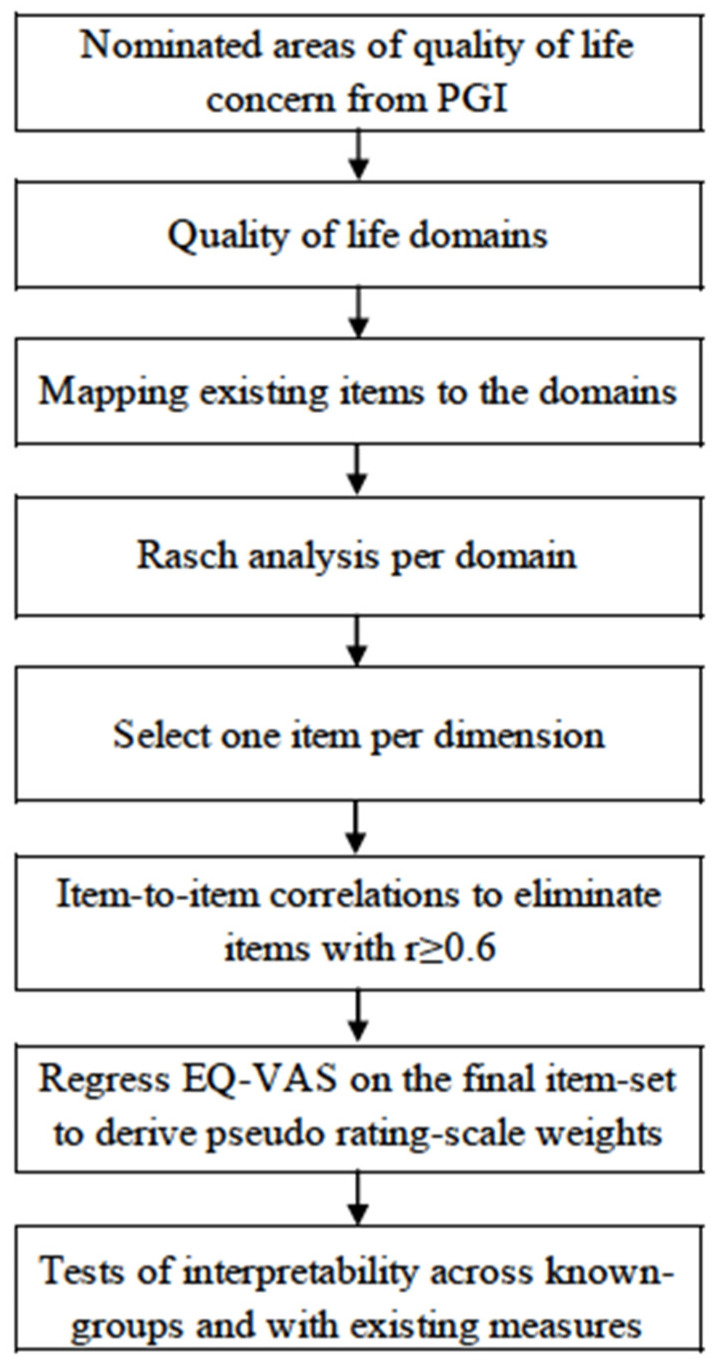
Steps to developing HIV-HRQL measure.

**Figure 2 jpm-12-02080-f002:**
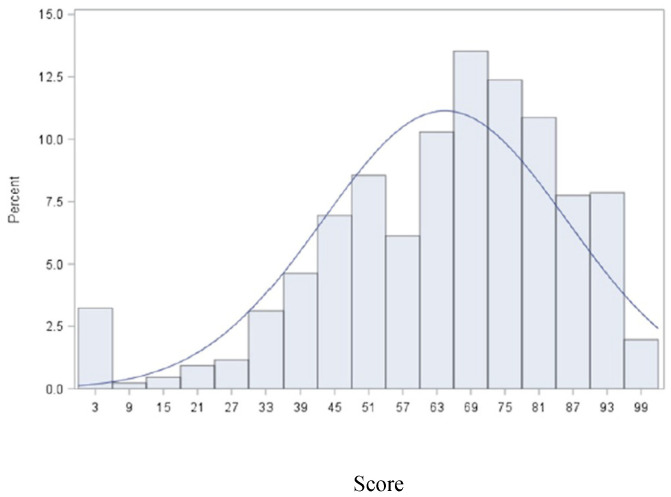
Distribution of PB-HIV Index score in people living with HIV.

**Table 1 jpm-12-02080-t001:** Characteristics of the Positive Brain Health Now cohort at study entry (N = 865).

Characteristics	Mean (SD) or N [%]	
	Men	Women
Participants	728 [84]	136 [16]
Age (years)	53.4 (8.3)	50.7 (7.4)
Education Primary school High school College/vocational diploma Bachelor MSc, PhD, or professional degree	28 [4] 179 [25] 238 [34] 180 [26] 76 [11]	11 [9] 47 [37] 46 [36] 15 [12] 7 [6]
Years since HIV	17.1 (8.1)	15.5 (6.7)
Time of diagnosis (1996) Pre cART Post cART	355 [49] 373 [51]	54 [40] 82 [60]
Living situation Alone Spouse or partner Family member Friends Other Missing	345 [47] 231 [33] 32 [4] 68 [9] 29 [4] 23 [3]	51 [38] 28 [21] 35 [26] 4 [3] 9 [6] 9 [6]
Days did you spend in bed due to an illness	4.3 (12)	6.8 (29)
Nadir CD4 < 200 cells/µL/% < 200	212.1 (165.8)/52	241.1(184.5)/48

B-CAM: Brief-Cognitive Ability Measure. cART: combined Antiretroviral Therapy.

**Table 2 jpm-12-02080-t002:** Items (original wording) with unstandardized (Beta) coefficients and weights.

Item (Scale)	Betas (SE)	Weights
** Pain/discomfort (EQ-5D-3L) **		
I have no pain or discomfort	Referent	0
I have moderate pain or discomfort	9.4 (2.1)	9
I have extreme pain or discomfort	13.9 (2.2)	14
** Fatigue Did you have a lot of energy? (RAND-36) **		
A little of the time/None of the time	Referent	0
Some of the time/A good bit of the time	4.2 (1.3)	4
Most of the time/All of the time	8.6 (1.5)	9
**Memory/concentration** **Miss appointments and meetings you had scheduled (PDQ-20)** ^a^		
Never	Referent	0
Rarely/Sometimes	4.8 (2.4)	5
Often/Almost always	7.3 (2.5)	7
**Physical appearance/Body image** Are you able to accept your bodily appearance? (WHOQOL BREF)		
Completely	Referent	0
Mostly/Moderately	3.9 (1.4)	4
A little/Not at all	5.9 (1.7)	6
** Sleep Do you feel rested when you wake up? (Sleep questionnaire) **		
Always	Referent	0
Often	2.3 (1.1)	2
Never	3.1 (1.8)	3
**Anxiety/Depression (EQ-5D-3L**)		
I am not anxious or depressed	Referent	0
I am moderately anxious or depressed	10.9 (2.2)	11
I am extremely anxious or depressed	14.6 (2.2)	15
** Motivation Do you have plans and goals for the future (Motivation questionnaire ** **)**		
A lot	Referent	0
Some	2.3 (1.5)	2
Not at all	3.2 (1.6)	3

^a^ PDQ-20 Perceived Deficit Questionnaire.

**Table 3 jpm-12-02080-t003:** Mean (SD) scores, correlations (95% Confidence Interval) for the PB-HIV Index and other measures of converging constructs.

	Mean (SD); Range (Higher Is Better)	Pearson’s Correlation (95% CI)
PB-HIV Index	64.4 (21.5); 0, 100	---
B-CAM	56.4 (14.4); 11.8, 97.1	0.14 (0.07, 0.21)
EQ-5D-3L Utility	0.8 (0.2); 0.2, 1	0.75 (0.72, 0.78)
SF-6D Index	0.7 (0.1); 0.3, 1	0.72 (0.69, 0.75)
Physical Component Summary score—Oblique	45.2 (10.3); 14.9, 61.5	0.72 (0.69, 0.75)
Physical Component Summary score—Orthogonal	46.7 (9.8); 14.1, 69.5	0.55 (0.51, 0.59)
Mental Component Summary score—Oblique	43.3 (12.4); 9.5, 67.2	0.74 (0.71, 0.77)
Mental Component Summary score—Orthogonal	43.9 (12.4); 11.8, 72.4	0.65 (0.61, 0.69)
WHO-QOL BREF		
Physical	68.5 (19.8); 0, 100	0.74 (0.71, 0.78)
Psychological	63.4 (18.5); 0, 100	0.78 (0.75, 0.81)
Level of Independence	69.4 (20.5); 6.3, 100	0.72 (0.68, 0.75)
Social Relationships	62.4 (20.3); 0, 100	0.51 (0.45, 0.55)
Environment	70.4 (16.6); 21.9, 100	0.61 (0.56, 0.65)
Spirituality/Religion/Personal Beliefs	70.5 (19.2); 7.3, 100	0.53 (0.47, 0.57)

B-CAM: Brief-Cognitive Ability Measure; SF-6D: Short Form—6 Dimensions; EQ-5D-3L Utility: EuroQol-5 Dimensions, 3 Levels Utility.

**Table 4 jpm-12-02080-t004:** Item-to-item polychoric correlations between the dimensions of the PB-HIV Index.

	Pain	Fatigue	Memory	Self-Image	Sleep	Motivation	Depression
**Pain**	1						
**Fatigue**	0.41	1					
**Memory**	0.33	0.34	1				
**Self-image**	0.32	0.46	0.32	1			
**Sleep**	0.38	0.55	0.28	0.31	1		
**Motivation**	0.19	0.39	0.16	0.26	0.19	1	
**Depression**	0.38	0.48	0.29	0.35	0.37	0.29	1

**Table 5 jpm-12-02080-t005:** Odds ratio and 95% CI on self-report adherence to medication and EQ-VAS and PB-HIV Index.

Forget ART Medication	Odds Ratio	95% Confidence Interval
EQ-VAS	1.08	0.99, 1.18
PB-HIV Index	1.13	1.05, 1.21

**Table 6 jpm-12-02080-t006:** Performance of PB-HIV Index total score and other generic measures across sex, time of diagnosis, disease severity, and comorbidities.

	N	PB-HIV Index	SF Index	EQ Utility
** Sex **				
Men	728	65.2 (20.7)	69.4 (12.4)	81.5 (16.4)
Women	136	60.5 (24.5)	69.2 (12.5)	80.8 (17.4)
** Age (years) **				
<45	136	65.3 (22.1)	69.6 (12.4)	82.1 (16.2)
45 to 55	405	62.6 (21.7)	68.5 (12.1)	80.4 (17.2)
55 to 65	252	64.2 (21.3)	69.5 (12.7)	81.3 (16.1)
>65	72	73.3 (17.5)	73.7 (12.1)	85.9 (13.7)
** Time of diagnosis **				
Before 1996	410	62.6 (21.9)	68.3 (12.6)	80.2 (16.9)
After 1996	455	66.1 (21.1)	70.4 (12.1)	82.5 (16.1)
** CD4 **				
Nadir <200	457	63.8 (22.1)	68.9 (12.9)	80.8 (17.3)
≥200	408	65.1 (20.9)	69.9 (11.7)	82.1 (15.6)
** Number of Comorbidities **				
0	404	66.2 (22.4)	70.9 (12.3)	82.9 (15.8)
1	213	63.6 (23.1)	69.4 (12.6)	81.5 (17.2)
2	133	62.6 (19.1)	67.9 (12.5)	79.4 (16.6)
3	63	61.1 (18.8)	66.3 (12.1)	77.5 (19.1)
4	52	62.8 (15.8)	66.3 (11.5)	79.3 (14.9)

**Table 7 jpm-12-02080-t007:** Characteristics of the patient experts and healthcare professionals who participated in the cognitive interviews.

Variables	Patient Experts (n)	Healthcare Professionals (n)
Sex (women/men)	1/4	4/4
Age range	20–50	20–45
Ethnicity	Asian (1), African (1), European (2), Middle Eastern (1)	Asian (1), European (1), North American (4), Middle Eastern (1)
Time since HIV diagnosis	>5 years (3), <2 years (2)	---
Languages	Bilingual (English and French n = 3) English only (n = 2)	Bilingual (English and French n = 7) English only (n = 1)
Education Bachelor Master PhD or Professional	1 2 1	2 2 4

**Table 8 jpm-12-02080-t008:** PB-HIV Index with seven dimensions with three declarative statements in English and French.

English	French
Select the option that best represents your health status in the last month	Sélectionnez l’option qui représente le mieux votre état de santé au cours du dernier mois
** Pain/discomfort **	** Douleur/inconfort **
I rarely have pain or discomfort	J’ai rarement de la douleur ou de l’inconfort
I have pain or discomfort some days	J’ai de la douleur ou de l’inconfort certains jours
I have pain or discomfort most days	J’ai de la douleur ou de l’inconfort la plupart des jours
** Fatigue **	** Fatigué **
I am rarely tired	Je suis rarement fatigué(e)
I am tired some of the days	Je suis fatigué(e) certains jours
I am tired most of the days	Je suis fatigué(e) la plupart des jours
** Memory/concentration **	** Mémoire/concentration **
I have no memory or concentration difficulties	Je n’ai pas de troubles de mémoire ou de concentration
My memory or concentration difficulties sometimes affects my daily life	Mes troubles de mémoire ou de concentration interfèrent parfois avec ma vie quotidienne
My memory or concentration difficulties frequently affects my daily life	Mes troubles de mémoire ou de concentration interfèrent fréquemment avec ma vie quotidienne
** Physical appearance/Body image **	** Apparence Physique/l’image corporelle **
I am satisfied with the way I look	Je suis satisfait(e) de mon apparence physique
I am somewhat satisfied with the way I look	Je suis plus ou moins satisfait(e) de mon apparence physique
I am not satisfied with the way I look	Je ne suis pas satisfait(e) de mon apparence physique
** Sleep **	** Sommeil **
I feel well rested when I wake up on most days	Je me sens reposé(e) quand je me reveille la plupart des jours
I feel well rested when I wake up on some days	Je me sens reposé(e) quand je me reveille certains jours
I rarely feel well rested when I wake up	Je me sens rarement reposé(e) quand je me réveille
** Depression **	** Dépression **
I rarely feel depressed	Je me sens rarement déprimé
I feel depressed some of the days	Je me sens déprimé certains jours
I feel depressed most of the days	Je me sens déprimé la plupart des jours
** Motivation **	** Motivation **
I often plan or set goals for my future	Je planifie ou me fixe souvent des objectifs pour mon avenir
I sometimes plan or set goals for my future	Je planifie ou me fixe parfois des objectifs pour mon avenir
I rarely plan or set goals for my future	Je planifie ou me fixe rarement des objectifs pour mon avenir

## Data Availability

Not applicable.

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
