# Peer review of "Development of a Prototype for a Bilingual Patient-Reported Outcome Measure of the Important Health Aspects of Quality of Life in People Living with HIV: The Preference Based HIV Index (PB-HIV)"

_jpm, 2022, doi:10.3390/jpm12122080_

Round 1

Reviewer 1 Report

The article if well written and the results are clearly presented. I have only comments to improve the manuscript:

Line 52 - It's hard to follow the problem with current HRQL approaches. It'd be easier to understand what is meant by or if the authors can give examples of what "domains" or "items" consist of (work/school, relationship, etc.?)

Line 113 - There's a typo on flowchart "Rasch analysis per domain"

Line 161 - Sometimes EQ VAS is used, other times EQ-VAS. 

Line 214 - Are the distribution of these characteristics in the cohort comparable to the general PLWH population?

Line 227 - Please define EQ-VAS

Author Response

Thank you for your and the reviewers’ comments on our manuscript. The reviewers raised some important points, and we have done our best to address their concerns. Below is our point-by-point response to the reviewers’ comments. As requested, both marked and clean copies of the manuscript are submitted.

Thank you for your consideration.

Reviewer 2

1.      Line 52 - It's hard to follow the problem with current HRQL approaches. It'd be easier to understand what is meant by or if the authors can give examples of what "domains" or "items" consist of (work/school, relationship, etc.?) 

In this context, domains represent broad latent constructs. Items are single questions that map to the domains.  The one item that best reflected the domain (Rasch analysis) was chosen to represent the dimension as typically preference-based measures are referred to as multi-dimensional.

2.      Line 113 - There's a typo on flowchart "Rasch analysis per domain" 

 Thank you for noticing the error. This has been fixed.

3.      Line 161 - Sometimes EQ VAS is used, other times EQ-VAS.  

These errors are fixed.

4.      Line 214 - Are the distribution of these characteristics in the cohort comparable to the general PLWH population? 

The data for this analysis was obtained from the Positive Brain Health Now (https://brainhealthnow.org) cohort which selected the participants using a cohort multiple randomized control trial design. The strength of this cohort is that members are fully characterized on patient-centered outcomes based on a strong theoretical framework, the Wilson-Cleary model, and widely tested and used in the health outcomes research.  While all consent-based observational studies have a potential for selection bias, the +BHN cohort was able to estimate its direction and impact resulting in the recruitment of a more vulnerable sample of people.

Mayo, N.E., Brouillette, MJ., Fellows, L.K. et al. Understanding and optimizing brain health in HIV now: protocol for a longitudinal cohort study with multiple randomized controlled trials. BMC Neurol 16, 8 (2016)

5.      Line 227 - Please define EQ-VAS 

EQ-Visual Analog Scale (EQ-VAS), is a self-rated health that asks the participants to rate their overall health perception on vertical analog scale (VAS) where the endpoints are anchored as ‘the best health you can imagine’ and ‘the worst health you can imagine’. This text is highlighted in the paper. (https://euroqol.org/eq-5d-instruments/eq-5d-5l-about/)

Reviewer 2 Report

In this manuscript, Dr. Mayo and his colleagues developed a preference-based HIV Index to measure the critical health aspects of quality of life in HIV patients. The authors described the procedures for setting the HIV-HRQL measure and compared the results with other methods. This study tries to establish a health state classification method associated with HIV patients, but the description of the system is not clear enough. 

The main concerns:

1.     I don’t understand why the authors choose these seven factors (pain/discomfort, fatigue, memory/concentration, physical appearance/body image, sleep, depression, and motivation) as dimensions. It would be clear for the authors to explain the theoretical basis.

Other concerns:

2.     In table 1, adding % in the [] would be better. If I don’t see the characteristics, I think the number in the [] means the reference. 

3.     How do you calculate the PB-HIV index, SF index, and EQ utility?

4.     Why do you show table 7? Do you think the factors such as sex, ethnicity, and language of the patient experts or healthcare professionals affect the outcome of the cognitive interviews? 

5.     Figure 2 does not have the X-axis and figure legend.

6.     It would be better to explain the meaning of the results rather than simply describe them.

7.     There are some grammar mistakes.

Author Response

Thank you for your and the reviewers’ comments on our manuscript. The reviewers raised some important points, and we have done our best to address their concerns. Below is our point-by-point response to the reviewers’ comments. As requested, both marked and clean copies of the manuscript are submitted.

Thank you for your consideration.

Reviewer 1

1.      I don’t understand why the authors choose these seven factors (pain/discomfort, fatigue, memory/concentration, physical appearance/body image, sleep, depression, and motivation) as dimensions. It would be clear for the authors to explain the theoretical basis. 

We started with the domains generated from the PGI and chose items that mapped to these domains and other domains. The total number of domains considered were 16.  As preference-based measures are based on independent dimensions (or minimally overlapping dimensions), we conducted interitem correlations and excluded dimensions with correlations ≥0.6) and those that were not actionable by the medical care.  For example, items representing feeling worried from HADS, and Depression from EQ-5D-3L correlated at 0.65 and only depression was retained in the final version.

Figure 1 and the methods in Step 5 and 6 provides relevant details.

2.      In table 1, adding % in the [] would be better. If I don’t see the characteristics, I think the number in the [] means the reference.  

Fixed

3.      How do you calculate the PB-HIV index, SF index, and EQ utility? 

We calculated the PB-HIV Index ourselves.  SF-6D Index and EQ-5D-3L Utility values are calculated using the standard published formulae for which the references are included.
Feeny, D., et al., Multiattribute and single-attribute utility functions for the health utilities index mark 3 system. Medical care, 2002. 40(2): p. 113-128.

Balestroni, G. and G. Bertolotti, EuroQol-5D (EQ-5D): an instrument for measuring quality of life. Monaldi Archives for Chest Disease, 2012. 78(3).

4.      Why do you show table 7? Do you think the factors such as sex, ethnicity, and language of the patient experts or healthcare professionals affect the outcome of the cognitive interviews?  

We chose to present these characteristics to show that our sample was diverse.

5.      Figure 2 does not have the X-axis and figure legend.  

Fixed

6.      It would be better to explain the meaning of the results rather than simply describe them. 

We have modified the discussion to explain better the meaning of the results.

7.      There are some grammar mistakes. 

      Fixed

Round 2

Reviewer 2 Report

The current version is much improved. I do not have further questions.